# The Effects of Intraguild Predation on Phytoplankton Assemblage Composition and Diversity: A Mesocosm Experiment

**DOI:** 10.3390/biology12040578

**Published:** 2023-04-10

**Authors:** Jun Da, Yilong Xi, Yunshan Cheng, Hu He, Yanru Liu, Huabing Li, Qinglong L. Wu

**Affiliations:** 1School of Ecology and Environment, Anhui Normal University, Wuhu 050031, China; 2State Key Laboratory of Lake Science and Environment, Nanjing Institute of Geography and Limnology, Chinese Academy of Sciences, Nanjing 210008, China; 3School of Life Sciences, Hebei University, Baoding 071002, China; 4Center for Evolution and Conservation Biology, Southern Marine Sciences and Engineering Guangdong Laboratory (Guangzhou), Guangzhou 511458, China

**Keywords:** Intraguild predation, environmental DNA, fish, shrimp, phytoplankton, diversity

## Abstract

**Simple Summary:**

The effects of the presence of predators (fish and shrimp) on the assemblage composition and diversity of phytoplankton were accurately characterized using outdoor mesocosms containing natural phytoplankton and zooplankton communities coupled with high-throughput sequencing. The results showed that the alpha diversity of phytoplankton and the relative abundance of Chlorophyceae increased with the addition of the fish *Pelteobagrus fulvidraco*, while the former decreased and the latter increased with the addition of the shrimp *Exopalaemon modestus*. The alpha diversity of phytoplankton and the relative abundance of Chlorophycea increased with the addition of fish and shrimp together, but these changes were less than the sum of the individual impacts of these two predators.

**Abstract:**

Intraguild predation (IGP) can have a significant impact on phytoplankton biomass, but its effects on their diversity and assemblage composition are not well understood. In this study, we constructed an IGP model based on the common three-trophic food chain of “fish (or shrimp)–large branchiopods (*Daphnia*)–phytoplankton”, and investigated the effects of IGP on phytoplankton assemblage composition and diversity in outdoor mesocosms using environmental DNA high-throughput sequencing. Our results indicated that the alpha diversities (number of amplicon sequence variants and Faith’s phylogenetic diversity) of phytoplankton and the relative abundance of Chlorophyceae increased with the addition of *Pelteobagrus fulvidraco*, while similar trends were found in alpha diversities but with a decrease in the relative abundance of Chlorophyceae in the *Exopalaemon modestus* treatment. When both predators were added to the community, the strength of collective cascading effects on phytoplankton alpha diversities and assemblage composition were weaker than the sum of the individual predator effects. Network analysis further showed that this IGP effect also decreased the strength of collective cascading effects in reducing the complexity and stability of the phytoplankton assemblages. These findings contribute to a better understanding of the mechanisms underlying the impacts of IGP on lake biodiversity, and provide further knowledge relevant to lake management and conservation.

## 1. Introduction

Intraguild predation (IGP) is a complex interspecific relationship prevalent in ecosystems, where two predators of the same trophic level compete for prey resources and may also act as predators of each other [1]. Intraguild predation is prevalent in various ecosystems [2,3]; for example, some small fish and shrimp in lake ecosystems are in competition for food resources such as zooplankton and also have predatory relationships [4]. Theoretical studies have shown that IGP can weaken the strength of trophic cascades in food webs and affect the biomass of lower trophic organisms. However, there are relatively few studies on the effects of IGP on the composition and diversity of lower trophic assemblages, and the related mechanisms need further research.

In lake ecosystems, IGP may affect the composition and diversity of phytoplankton assemblage through both top-down and bottom-up effects. As a link between predators and primary producers, zooplankton community composition is a sensitive indicator of food web structure and can reflect the strength of IGP cascading down to the phytoplankton trophic level [5,6]. Top-down control by predators can shift the size distribution of zooplankton communities to smaller species, which exhibit relatively less feeding pressure on phytoplankton, thereby increasing the likelihood of phytoplankton blooms [7]. Meanwhile, small zooplankton had higher prey selectivity than large zooplankton [8], which will have an impact on the assemblage composition and diversity of phytoplankton [9,10]. Alternatively, IGP-induced changes in the composition of the zooplankton community can also strongly affect the composition and diversity of phytoplankton assemblage through trophic cascade and nutrient uptake, known as “bottom-up effects” [11,12,13]. Specifically, alterations in the density and composition of zooplankton communities resulting from IGP can impact the concentration of nutrients, such as nitrogen and phosphorus, in the water column, altering the nutrient availability for phytoplankton and ultimately affecting their assemblage composition and diversity. For example, small-sized (<1.2 mm) zooplankton communities have a higher nutrient recycling efficiency than do their larger (>1.2 mm) counterparts [14,15], and communities dominated by small-sized zooplankton at equal biomass have a higher nutrient recycling rate than those dominated by larger-sized zooplankton [16]. Moreover, large cladocerans have higher phosphorus but lower nitrogen requirements than do small cladocerans and copepods, which further affects the nutrient availability and leads to changes in phytoplankton assemblage diversity through bottom-up effects [17]. Fish (or shrimp) also directly release nutrients into the water through excretion and defecation, which in turn affects water nutrient concentration and subsequently the composition and diversity of phytoplankton assemblages [11]. However, there is relatively little research on how IGP affects the composition and diversity of phytoplankton assemblages, and further investigation is required to determine the underlying mechanisms.

In previous studies, the impact of IGP on phytoplankton has mainly been based on the observation and identification of phytoplankton species under a microscope [4,18,19]. However, the resolution of species identification using microscopy is limited [17], and it cannot provide information on genetic diversity. In addition, microscopy-based methods are restricted to phytoplankton with sufficient morphological features and may also misidentify or overlook fragile and non-fixable phytoplankton [20], or small phytoplankton that cannot be accurately distinguished by conventional light microscopy [21]. It should be noted that while microscopy-based methods are valuable for assessing phytoplankton diversity, they may not be able to provide a comprehensive characterization, especially phylogenetic diversity, of the entire assemblages [22]. Currently, high-throughput sequencing of environmental DNA (eDNA) has matured as a technology and has been used for monitoring biodiversity in lake ecosystems [23]. Compared with traditional methods of microscopy-based species identification, eDNA high-throughput sequencing has the advantages of high sensitivity; standardized methods; and low human, material, and time costs, making it possible to discover the vast diversity of phytoplankton in lake ecosystems, including microorganisms, and rare and fragile species [21,24]. This technology can provide convenient and reliable technical support for studying the impact of IGP on the composition and diversity of phytoplankton assemblages.

The present study is based on a three-level trophic chain commonly found in subtropical shallow lakes in China, consisting of "fish (shrimp)–large cladocerans (*Daphnia pulex*; Leydig, 1860)–phytoplankton." A within-group predation model was constructed using a mesocosm experiment to investigate the impact and mechanisms of IGP on the composition and diversity of phytoplankton assemblages, which were explored using eDNA high-throughput sequencing. The results will provide new knowledge relevant to the management and regulation of aquatic ecosystems in subtropical shallow lakes. 

## 2. Materials and Methods

### 2.1. Experimental Design

The mesocosm experiment of this study was conducted at the Taihu Lake Ecosystem Research Station of the Chinese Academy of Sciences (31°24′13″ N, 120°13′56″ E), located in Meiliang Bay on the northern shoreside of Lake Taihu, China. The experimental system consisted of 24,500-L plastic tanks, each with a height of 97 cm, an inside diameter of 90 cm at the top, and an inside diameter of 78 cm at the base. The tanks were filled with 480 L of water from eutrophic Lake Taihu, which was pre-screened using a 64 μm mesh to remove crustacean zooplankton and large inorganic particles. The water was mixed thoroughly to ensure homogeneity before it was added to the tanks. Then, 100 *D. pulex* individuals (average body length of 1.6 mm) were added to each tank as the main filter-feeding zooplankton, at a density of 0.2 individuals L^−1^, which is similar to the spring *Daphnia* density in Lake Taihu [25]. To prevent entry by other predators such as aquatic insects, each mesocosm was covered with mesh screens with a mesh size of 0.83 mm and a light transmittance of 77% [4].

Following a semi-confined incubation period of two weeks in experimental tanks prior to the commencement of the study on 7 April 2020, a zooplankton community consisting predominantly of *Daphnia* (constituting over 90% of the biomass) and a phytoplankton assemblage dominated by diatoms (constituting over 80% of the biomass) were established in each tank. Notably, no significant differences were observed in the biomass or composition of the plankton communities across the experimental tanks. We have selected two common predators found in a subtropical lake in China: the small-sized fish species *Pelteobagrus fulvidraco* (Richardson, 1846) and the shrimp *Exopalaemon modestus* (Heller, 1862). Gut content analysis (GCA) studies have revealed that *P. fulvidraco* preys on *E. modestus* [26,27], while both predators consume zooplankton as part of their diet. The experiment consisted of four treatments: C treatment (control with no predators added), F treatment (added *P. fulvidraco*), S treatment (added *E. modestus*), and FS treatment (added both *P. fulvidraco* and *E. modestus*). Each treatment had six replicates. The mesocosms for different treatments were arranged in alternating positions. Each F treatment contained two *P. fulvidraco*, with a biomass of 4.5 ± 0.5 g. The fish were obtained from local aquaculture around Taihu Lake and were acclimatized for one week before being added to the experimental tanks. The number of shrimp added was determined based on the natural sex ratio during the breeding season, which included 18 females (0.5 ± 0.15 g tail^−1^) and 12 males (0.16 ± 0.08 g tail^−1^) [27]. The shrimp were harvested from Taihu Lake using cages. Nutrient salts (elemental phosphorus: 5 (μg L^−1^ day^−1^); elemental nitrogen: 130 (μg L^−1^ day^−1^) were added daily to simulate the exogenous nutrient salt load of Taihu Lake [28].

### 2.2. Sampling and Processing

Sampling was taken on the 84th day of the experiment (at approximately 8:00 a.m.), when *E. modestus* had reproduced for one generation and reached the adult stage. To minimize spatial heterogeneity, the water in each experimental system was mixed separately prior to sampling, and 1 L of water was collected from approximately 0.5 m below the water surface using a water collector. The water samples were stored in sterilized polyethylene bottles and refrigerated at 4 °C for transportation to the laboratory. Phytoplankton eDNA samples were collected from the water samples upon arrival at the laboratory. Additionally, a total of 11 L of water samples were collected separately for the determination of total nitrogen (TN), dissolved total nitrogen (DTN), total phosphorus (TP), dissolved total phosphorous (DTP), and chlorophyll *a* (Chl-*a*) using methods specified in the Specification for Lake Eutrophication Investigation [29]. The remaining 10 L of water samples were filtered through a 64-μm plankton net, and zooplankton were collected into 50 mL white bottles, preserved by adding 2 mL Lugol reagent, then identified and counted under a 40× microscope, and the biomass (dry weight) of each zooplankton species was estimated based on its length–body weight relationship [30].

### 2.3. High-Throughput Sequencing

The water samples were collected in polyethylene bottles and then filtered through a 0.2-µm polycarbonate membrane (Millipore, Burlington, MA, USA) [31] using negative pressure (<20 mbar) to collect phytoplankton eDNA samples. The filter membranes were stored at −80 °C in an ultra-low temperature freezer prior to genomic DNA extraction. Genomic DNA was extracted from phytoplankton eDNA samples using an improved phenol–chloroform extraction and ethanol precipitation method [31]. The 18S rRNA gene of all samples was amplified by polymerase chain reaction (PCR) using the EK-NSF573 5′-CGCGGTAATTCCAGCTCCA-3′ and Ek-NSR951 5′-TTGGYRAATGCTTTCGC-3′ primers [32], with a fragment length of approximately 560 bp [33]. High-throughput sequencing was conducted using the Illumina Miseq platform [23]. The PCR amplification reaction system consisted of a total volume of 25 μL, which included the following: 0.1 μL (20 pmol L^−1^) of the forward primer and 0.1 μL (20 pmol L^−1^) of the reverse primer, 3~5 μL of DNA template, 12.5 μL of 10× AccuPrime^TM^ PCR buffer II, 0.5 μL of AccuPrime^TM^ Taq high-fidelity DNA polymerase (Invitrogen, Carlsbad, CA, USA), and sterile deionized water added to a final volume of 25 μL. The PCR amplification conditions consisted of an initial denaturation step at 94 °C for 5 min, followed by 30 cycles of denaturation at 94 °C for 50 s, annealing at 57 °C for 50 s, and extension at 72 °C for 1 min, and a final extension step at 72 °C for 10 min [32]. To minimize random errors in product generation, each sample was amplified by PCR three times. The PCR products from the three amplifications were then mixed and purified using the QIAquick Gel Extraction Kit (Qiagen, Germantown, MD, USA). After purifying, PhiX (Illumina, San Diego, CA, USA) was added to each sample before sequencing was performed using the Illumina MiSeq platform.

The raw data obtained from sequencing were trimmed using the TrimGalore software version 0.4.4 to remove the primer and adapter sequences from the ends, and then the Fastx software version 0.0.13 was used to remove bases with a quality score of less than Q15 from the ends of the sequences. The data were merged using the FLASH2 software version 1.2.7 to obtain effective sequences. The USEARCH software version 7.1 was then used to discard sequences with more than two mismatches in the primers, a length of less than 100 bp, and a total base error rate greater than two, resulting in optimized sequences [32]. Finally, the sequences were clustered into Amplicon Sequence Variants (ASVs) based on a specified similarity threshold using the UPARSE software version 7.0.1001. The similarity threshold for ASVs in each sample was set at 99%. The clustered ASVs were annotated using the Silva123 database. ASVs that appeared only once in all samples were removed, the total number of sequences was calculated for each sample, and the samples were rarefied to the minimum number of sequences for subsequent data analysis [34]. The alpha diversity of phytoplankton in each sample was characterized using the number of ASVs [35] and Faith’s phylogenetic diversity (Faith’s PD) [36].

### 2.4. Statistics Analyses

SPSS software version 20.0 was used to perform one-way ANOVA to analyze differences in environmental indicators, the biomass of various zooplankton groups, Chl-*a*, and the alpha diversity of phytoplankton among different treatment systems [23]. The glm function in the glmmTMB v1.0.2.1 package in R 4.1.2 was used to conduct generalized linear mixed model analysis in studying the effects of fish, shrimp, and their combined effects on trophic cascades [4]. For each response variable, we evaluated the relative fit of four candidate models (fixed effect: fish × shrimp, fish, shrimp, and null) and selected the best model based on the lowest AIC value using the default ANOVA function. If the interaction term was significant in the best model, we conducted a post hoc test on the best model to examine potential antagonistic or synergistic effects between both predators, using Tukey’s post hoc tests (function "emmeans" from the "emmeans" package). The vegan v2.5.7 package in R 4.1.2 was used to conduct principal coordinate analysis (PCoA) based on the Bray–Curtis distance on the composition of phytoplankton assemblages, and ADONIS and ANOSIM were used to analyze the differences in phytoplankton assemblage compositions among different experimental treatment systems [37]. Canonical correspondence analysis (CCA) was used to analyze the correlation between environmental factors and changes in the composition of phytoplankton assemblages. In addition, multiple linear regression analysis and structural equation modeling (SEM) were used to explore the relative contributions of various environmental factors to the changes in alpha diversity of phytoplankton and the direct and indirect effects of fish and shrimp on the composition of phytoplankton assemblages (PCoA first axis) [38]. Both of these effects were calculated using standardized path coefficients (multivariate regression coefficients estimated by maximum likelihood) in the lavaan v0.6.3 package in R 4.1.2 [39].

Molecular ecological network analysis was used to investigate the effects of within-group predation on the interactions between phytoplankton. To avoid spurious correlations, only ASVs present in more than 50% of the samples were selected for Spearman correlation analysis, and ASVs with |r| > 0.60 and *p* < 0.05 were screened out [40]. The average clustering coefficient, modularity index, average degree, and other network topological parameters were calculated using cytoscape version 3.5.1. In addition, 1000 random Erdos–Renyi networks were generated to calculate their topological parameters, and z-tests were used in R to test for differences in topological features between the observed network and the random networks [41].

## 3. Results

### 3.1. Environmental Factors

At the end of the experiment, the addition of *P. fulvidraco* and *E. modestus* had different effects on the nutrient concentrations in the experimental tanks. One-way ANOVA analysis showed that, compared to the C treatment, the concentrations of DTN in the treatments with fish, shrimp, and fish and shrimp combined had all significantly decreased (*p* < 0.05; Figure 1b), but there was no significant difference in DTN concentrations among the three treatments (*p*  > 0.05; Figure 1b). In contrast, the trend for TP concentration was the opposite to that of DTN (Figure 1c). The TN concentrations in all three predator addition treatments were lower than that in the control (Figure 1a), with the TN concentration in the F treatment (1.92 mg L^−1^) significantly lower than in the C treatment (2.56 mg L^−1^) (*p* < 0.05; Figure 1a). The DTP concentration in the S treatment was the lowest (at 0.021 mg L^−1^), which was significantly lower than that in the F treatment (0.025 mg L^−1^) and in the FS treatment (0.028 mg L^−1^) (*p*  < 0.05; Figure 1d).

After performing generalized linear mixed model analysis, we found a significant interaction effect of fish and shrimp on the TN, TP, and Chl-*a* concentrations, and *D. pulex* biomass (Table 1). Specifically, the addition of fish and shrimp led to a significant reduction in TN and *D. pulex* biomass, and a significant increase in TP and Chl-*a*. The interaction effect of fish and shrimp showed a significant reduction in TP and Chl-*a*, but a significant increase in TN and *D. pulex* biomass. These results suggest that the addition of *P. fulvidraco* and *E. modestus* had complex impacts on the aquatic ecosystem, affecting multiple trophic levels and nutrient cycles.

### 3.2. Trophic Cascade

The addition of *P. fulvidraco* and *E. modestus* had different effects on the biomass of low trophic level organisms (Chl-*a* and zooplankton) in the treatments. The total zooplankton biomass in the F and S treatments was significantly different from each other at the end of the experiment (*p*  < 0.05; Figure 2a), but there was no significant difference between F and FS or S and FS treatments (*p* > 0.05; Figure 2a), and all three were lower than the C treatment.

The biomass of different zooplankton groups also showed significant changes in the presence of fish and shrimp (Figure 2b–e). Compared to the C treatment, the biomass of copepods in the F treatment increased significantly (*p*  < 0.05; Figure 2b), and the biomass of small branchiopods and rotifers also increased (Figure 2c,d), while *Daphnia* species were not present (Figure 2e). The biomass of copepods and small branchiopods in the S treatment decreased (Figure 2b,c), whereas the biomass of rotifers increased significantly (*p*  < 0.05; Figure 2d), and *Daphnia* were not present as per the F treatment (Figure 2e). The biomass of copepods decreased in the FS treatment (Figure 2b), but those of small branchiopods and rotifers increased, and *D. pulex* was present, though only in two out of six experimental systems, with an average biomass of 13.2 mg L^−1^ and thus much lower than that of the C treatment (357.8 mg L^−1^) (Figure 2c–e). Compared to the C treatment, the Chl-*a* of all three predation treatments increased significantly (*p* < 0.05), and the increase in the FS treatment was smaller than that in the F treatment and S treatment (Figure 2f).

### 3.3. Diversity and Assemblage Composition of Phytoplankton

A total of 173,022 high-quality sequences related to phytoplankton were obtained after splicing, filtering, and ‘de-chimerizing’ the raw data of the 24 phytoplankton eDNA samples obtained through high-throughput sequencing using 18S rDNA genes, with an average length of 560 bp. The remaining high-quality reads of all these samples were resampled to the minimum number (20,464), and 464 ASVs (99%) were obtained. Phytoplankton alpha diversity, as measured by the number of ASVs and Faith’s PD, differed significantly among the treatments. The alpha diversity increased significantly in the F treatment but decreased significantly in the S and FS treatments compared to the C treatment (one-way ANOVA, all *p* < 0.05, Figure 3a,b).

Among the detected 464 ASVs, 336 ASVs belonged to Chlorophyta, accounting for 93.9% of the total number of sequences (Figure 3d). Chlorophyta was mainly dominated by Chlorophyceae, followed by Trebouxiophyceae and Ulvophyceae, Cryptophyta, Pyrrophyta, and Ochrophyta. The relative abundance of Cryptophyta was higher in the F treatment compared to the C treatment, while Trebouxiophyceae was much higher in the C treatment (Figure 3e). The relative abundance of Trebouxiophyceae in the F, S, and FS treatments was only 3.6%, 0.8%, and 0.7%, respectively (Figure 3e). Chlorophyceae was the dominant taxon (12.4%) in the C treatment, but was much more abundant in the F, S, and FS treatments (80.1%, 94.9%, and 97.6%, respectively; Figure 3e). The relative abundance of Cryptophyceae was higher in the F treatment (14.5%), and the relative abundance of Eustigmatophyceae was higher in the S treatment (3.9%) (Figure 3e). Chrysophyceae was detected in the C treatment, with a relative abundance of 2.7%, and brown Raphidophyceae was only detected in the F treatment (Figure 3e).

Principal coordinate analysis based on Bray–Curtis distance was used to investigate differences in the phytoplankton assemblage composition between treatments. The result showed that the first three principal coordinates contributed 32.1%, 18.6%, and 10.8%, respectively, of the total variance. The PCoA plot of phytoplankton assemblage composition indicated that the taxonomic composition of phytoplankton in the different treatments aligned along PCoA axis 1 and then axis 2 (Figure 3c). The ADONIS and ANOSIM tests further found that the phytoplankton assemblage composition in the C treatment was significantly different from that in the other three treatments (*p* < 0.05; Table 2). In addition, the phytoplankton assemblage composition in the F treatment differed significantly (*p* < 0.05; Table 2) from that in the S and FS treatments. However, the phytoplankton assemblage composition in the S treatment did not differ significantly from that in the S treatment (*p* > 0.05; Table 2).

### 3.4. Interactions among Phytoplankton

There were significant differences in the interactions among phytoplankton between treatments (Figure 4, Table 3). Comparative analysis of four different molecular networks that were constructed based on the ASVs of phytoplankton assemblages of each treatment showed that the topological characteristics of these four networks were different from their random networks, indicating that the molecular networks of phytoplankton in each treatment were non-random (Figure 4, Table 3). The phytoplankton molecular network in the C treatment had the most nodes (191) followed by those in F (143) and FS (134) treatments, whereas the S treatment had the least nodes (57). In contrast, the differences in the phytoplankton molecular network edges among the treatments were mainly characterized by the highest being in the S treatment (1966), the second highest in the FS treatment (1820), and the least being in the F treatment (711), while the number of network edges in the C treatment (1612) was between those of the F and FS treatments (Figure 4, Table 3). This indicates that the phytoplankton assemblages in the S treatment had more complex interconnections and network structures. The modularity, mean clustering coefficient, mean path length, and diameter of the molecular network of the phytoplankton assemblages were highest in the C treatment, while these parameters were lower in the FS treatment than in either the F or S treatments, indicating that the complexity and stability of the molecular network of the phytoplankton in the C treatment were highest, followed by the FS treatment, whereas the lowest was in the F and the S treatments (Table 3, Figure 4). Compared to the C treatment, the proportion of negative correlations in the phytoplankton molecular network was significantly higher in the F treatment, while it was substantially lower in the S and FS treatments (only 19.4% and 23.1%, respectively), while the trend of positive correlations was the opposite (Figure 4, Table 3).

### 3.5. Environmental Factors Affecting Phytoplankton Diversity and Assemblages Composition

A multiple linear regression model showed that copepod biomass had a significant positive effect on both phytoplankton ASV number and PD values, accounting for a large proportion of the variation (32.2% and 31.8%, respectively; Figure 5a,b). Other variables with significant effects but smaller importance included TN and the biomass of small branchial hornworms, which had positive effects, and TP, TDN, TDP, the biomass of *Daphnia*, and rotifers, all of which had negative effects. The standardized regression coefficients and their *p* values revealed that the positive effect of copepod biomass on phytoplankton ASV and PD values was particularly strong (Figure 5a,b). Meanwhile, the effects of other variables were relatively small or not significant for phytoplankton PD values. The negative effects of some variables, such as TP, TDN, and *D. magna* biomass, suggest that these factors may limit the growth of phytoplankton in aquatic ecosystems.

According to the results of Detrended Correspondence Analysis (DCA), both the first and second axes of the sample matrix were greater than two, indicating a nonlinear relationship between species and environment, therefore CCA was chosen to assess the correlation between environmental factors and changes in phytoplankton assemblage composition. The results showed that TN, TP, DTN, and the biomasses of copepods, large branchial hornworms, and rotifers were significantly correlated with changes in the phytoplankton assemblage composition (*p* < 0.05, Figure 5c). The first two CCA axes together explained 24.69% of the variation in the phytoplankton assemblages (Figure 5c), and all six axes explained 39.13%. The first axis was highly correlated with TN, TP, DTN, copepod biomass, and large branchial hornworm biomass, with R^2^ values of 0.54, 0.84, 0.64, 0.69, and 0.77, respectively. The second axis was highly correlated with rotifers biomass, with an R^2^ value of 0.36. Furthermore, the results of SEM analysis showed that rotifers biomass and total phosphorus concentration are the main direct factors affecting the changes in the phytoplankton assemblage composition, with a positive correlation between rotifers biomass and total phosphorus (Figure 6). The relationship between large branchial hornworm biomass and phytoplankton assemblage composition was not statistically significant. Fish and shrimp indirectly influence the phytoplankton assemblage composition through their consumption of zooplankton and nutrients. The addition of fish and shrimp predators showed a negative correlation with large branchial hornworm biomass, while large branchial hornworm biomass showed a negative correlation with rotifers biomass and total phosphorus, and a positive correlation with phytoplankton assemblage composition (Figure 6).

## 4. Discussion

### 4.1. Cascade Impact on Nutrient Cascades

The vast majority of phytoplankton suppression operations are encouraging zooplankton to lower trophic levels, producing a significant cascade effect [42,43,44]. Both *P. fulvidraco* and *E. modestus* are important natural predatory enemies of zooplankton in subtropical lakes [26,45], and in agreement with theoretical and empirical studies of linear food chains [2], predation on zooplankton (especially *Daphnia*) by *P. fulvidraco* and *E. modestus* diminishes zooplankton grazing pressure on phytoplankton but promotes phytoplankton diversity and composition through trophic cascade effects (Figure 2f) [4,46,47].

There are also differences in the predation preferences of zooplankton by *P. fulvidraco* and *E. modestus*. Previous studies have shown that fish preferentially prey on larger zooplankton and select *Daphnia* much more than copepods of the same body size [48,49]. Consistent with these findings, the present study observed a significant increase in copepod biomass in the F treatment and a significant increase in rotifers biomass in the S treatment (Figure 2b,d), with *Daphnia* disappearing from both treatments (Figure 2e). The rapid growth of rotifers in the S treatment was attributed to competition and predation by other zooplankton, as the size range of food particles edible for cladocerans covered the edible size range for rotifers [50], indicating that there is competition for food ecological niches between cladocerans and rotifers. Compared to the C treatment, the biomass of copepods and small branchiopods was reduced in the S treatment, allowing rotifers to benefit from competition and thus gain an advantage.

### 4.2. Intraguild Predation Impact on Nutrient Cascades

In this study, we investigated the intensity of the effect on zooplankton and the cascade effect on phytoplankton diversity and composition in the presence of *P. fulvidraco* and *E. modestus*, both separately and together. We found that a significant interaction effect of fish and shrimp on the nutrient cascades though the combined presence of both predators did not result in a stronger effect on zooplankton and phytoplankton than the presence of a single predator. Compared to the C treatment, the total zooplankton biomass was reduced and Chl-*a* was significantly higher in both the F and S treatments (Figure 2a,f). However, both the reduction and increase were smaller in the FS treatment than in the F and S treatments, indicating the existence of IGP between the two predators. The IGP intensity of predators on zooplankton was species-dependent [51,52], with the greatest intensity of predation on *Daphnia* by *P. fulvidraco* and *E. modestus* resulting in its disappearance in the F and S treatments. *Daphnia* were present in the FS treatment, but their biomass was much lower than in the C treatment (Figure 2e). This is consistent with the results of previous studies [4,12], which have shown that a strong IGP is usually found in the population diversity of *Daphnia*.

*Daphnia* is the main filter-feeding zooplankton, exerting relatively high feeding pressure on phytoplankton, thereby reducing the potential for phytoplankton blooms [7,53,54]. In our study, we found that the addition of *P. fulvidraco* and *E. modestus* significantly reduced the biomass of *Daphnia* and its dominance, and thus promoted phytoplankton diversity and composition (Figure 2e). However, the presence of IGP between fish and shrimp increased the survival rate of *Daphnia*, which in turn had an antagonistic effect on phytoplankton diversity and composition, as reflected by a smaller increase in Chl-*a* in the FS treatment than in the FS treatment.

The IGP between fish and shrimp also had an antagonistic effect on nutrient concentrations in the water column, with a significant increase in TP and decrease in TN in the experimental treatments relative to the C treatment (Figure 1a,c). However, the increasing trend of TP and the decreasing trends of TN and DTN in the F and S treatments were smaller than those in the FS treatment (Figure 1a−c). The predation of *P. fulvidraco* and *E. modestus* could affect the nutrient concentration in the water column by changing the composition of zooplankton [4]. Large branchiopods have higher phosphorus requirements but lower nitrogen requirements than small branchiopods and copepods [17,55], and the changes in the biomass of large branchiopods will inevitably have an impact on TN and TP concentrations in the water column [56]. Meanwhile *P. fulvidraco* and *E. modestus* can directly release nutrients into the water through excretion [11], and the predation of shrimp by fish leads to a decrease in overall nutrient excretion. According to the “consumer-driven nutrient cycling” theory of ecological chemometrics [57,58], *P. fulvidraco* and *E. modestus* ingest and use more nitrogen for their own growth and reproduction, and predation by fish and shrimp in the FS treatment led to a further decrease in nitrogen use by both predators than in the F and the S treatments. The results of the GLM in the present study support this view, and the interactive effects of fish and shrimp on TN and TP were significantly different to that of fish or shrimp alone (Table 1). It may be that *P. fulvidraco* and *E. modestus* favor phytoplankton diversity and composition, both downstream (reduction of zooplankton) and upstream (release of nutrient salts). However, the IGP between fish and shrimp is antagonistic in its effects on both zooplankton and nutrient concentrations, which in turn has a weaker cascade effect on phytoplankton than a single predator effect, with a smaller increasing trend in Chl-*a* than in the presence of a single predator.

### 4.3. The Presence of Shrimp Shapes Phytoplankton Diversity and Assemblage Composition through Top-Down and Bottom-Up Effects

Although in this study the cascade effects on phytoplankton in the FS treatment were weaker than in the F and S treatments, there was no corresponding difference in phytoplankton assemblages diversities (alpha and beta diversities) among these three treatments (Figure 3). Compared with the C treatment, phytoplankton alpha diversities increased in the F treatment. Previous studies have found that copepods can control large phytoplankton and can also relieve small phytoplankton from the grazing pressure caused by intermediate consumers (protists) [59,60]. The present study found a positive correlation between the alpha diversities of phytoplankton and copepod biomass, which may be the main reason for the increased diversity in the F treatment (Figure 5a,b). The grazing of copepods can potentially decrease the abundance of specific phytoplankton species, which will create opportunities for other species to thrive and ultimately result in increased phytoplankton diversity. This ecological phenomenon is commonly referred to as “grazing control” and has been observed in various marine ecosystems [61,62]. By contrast, phytoplankton alpha diversities in the S and FS treatments were significantly lower than in the C and F treatments, and their assemblage compositions were similar but significantly different from those of the C and F treatments (Figure 3a–c). The ecological effects of predators depend on their density [63,64]. For example, during the period of high abundance of mysis shrimp in late spring and early summer, the appearance of high-density predators led to the rapid growth of rotifers and a tendency towards a smaller zooplankton community size [49,65]. Rotifer biomass and nutrient concentration are the direct factors affecting changes in phytoplankton assemblage composition (Figure 6). In addition, as the dominance of small zooplankton increased, the compositions of phytoplankton assemblage in the treatments also changed. The most obvious change was the rapid dominance of Chlorophyceae (with a relative content of 94.9 and 97.6% in the S and FS treatments, respectively; Figure 3d). Some species of Chlorophyceae have a strong competitive ability for resources, and the nutrient cycling driven by fish and shrimp results in the dominance of algae with high absorption rates [66,67].

The diversity and assemblage composition of phytoplankton are also regulated by nutrient levels and higher trophic levels, through bottom-up effects (such as the effects of nutrients such as nitrogen and phosphorus) and top-down effects (such as zooplankton predation) effects [64,65,66,67]. Previous studies have shown that TP is one of the main factors limiting the primary productivity of phytoplankton [68], and *Daphnia* is a key zooplankton for manipulating phytoplankton [69]. The selective predation of small zooplankton such as rotifers also has a substantial impact on the phytoplankton assemblage composition [68,69]. Similarly, the result of SEM path analysis in the present study showed that rotifer biomass and TP were direct factors affecting changes in phytoplankton assemblage composition, while *Daphnia* was an indirect factor (Figure 6).

Intraguild predation also affected the complexity and stability of the phytoplankton assemblages. It is assumed biodiversity is positively related to the complexity and stability of an assemblage [70]. Thus, the lower diversities in the F and FS treatments indicate a simpler and less stable phytoplankton assemblage. This was further confirmed by our molecular network analysis, which showed that the networks in these two treatments had the highest average degree (Figure 6, Table 3), indicating positive connections between forming clusters [41]. In addition, the positive correlations in these two networks were nearly four times that of the negative correlation, which indicates that positive feedback loops supporting mutual adaptability between phytoplankton were created to disrupt assemblage stability [70].

## 5. Conclusions

In this study to investigate how IGP impact on phytoplankton diversity and assemblage composition, we constructed an IGP model based on the common three-trophic food chain of “fish (or shrimp)–large branchiopods (*Daphnia*)–phytoplankton” in outdoor mesocosms and applied environmental DNA high-throughput sequencing to perform a deep analysis of the phytoplankton 18S rRNA gene of diversity. The results showed that in single predator treatments, both predators induced a strong trophic cascade, as evidenced by remarkable changes in phytoplankton alpha diversities (the number of ASVs and Faith’s PD), the relative abundance of Chlorophyceae, and Chl-*a*. However, these changes were less than the sum of the individual impacts of these two predators, which indicated the strength of collective predator effects on phytoplankton. Network analysis further demonstrated that this IGP effect weakened the complexity and stability of the phytoplankton assemblages. These findings enhance our comprehension of the mechanisms behind the effects of IGP on lake biodiversity and provide further knowledge that is relevant to the management and conservation of lakes.

## Figures and Tables

**Figure 1 biology-12-00578-f001:**
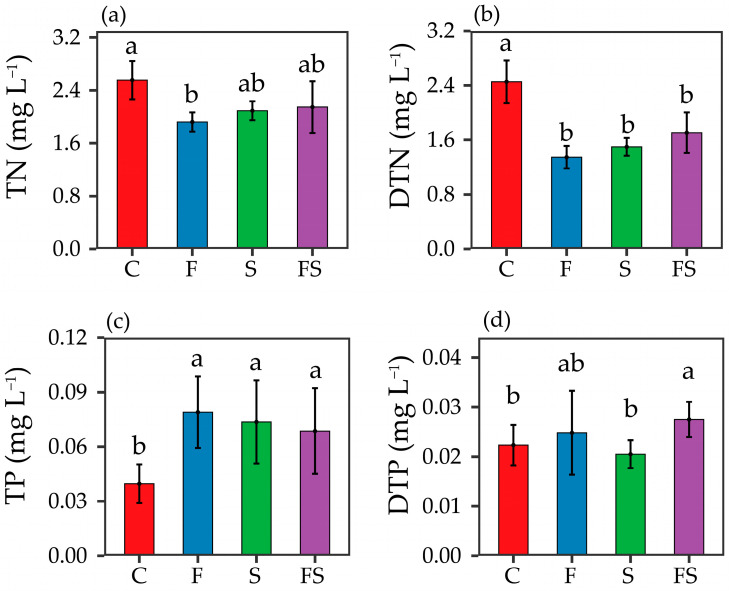
Differences in environmental indicators among treatments. (**a**) Total nitrogen (TN), (**b**) Dissolved total nitrogen (DTN), (**c**) Total phosphorus (TP), (**d**) Dissolved total phosphorous (DTP). C: Control, F: Fish-added treatment, S: Shrimp-added treatment, FS: Fish–shrimp-added treatment; each treatment was replicated six times. Different letters above the same environmental indicator bar indicate statistically significant differences according to the results of one-way ANOVA.

**Figure 2 biology-12-00578-f002:**
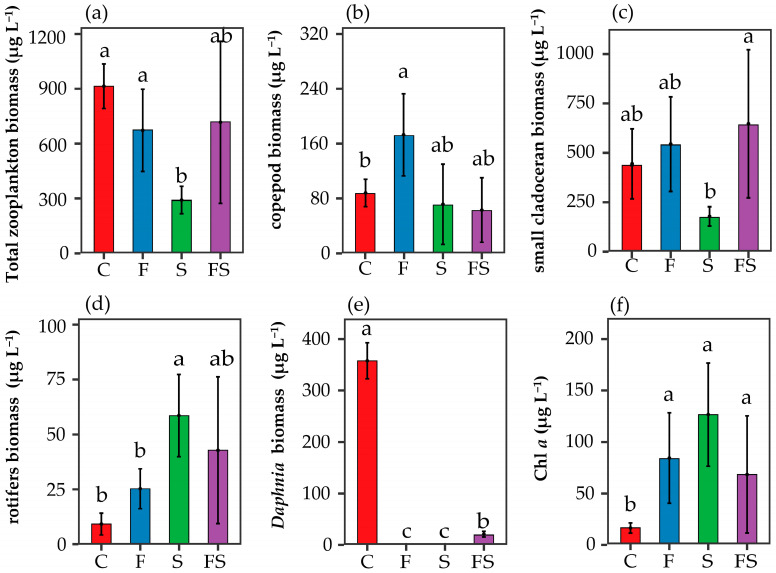
Trophic level changes under different predator scenarios. (**a**) Total zooplankton biomass, (**b**) copepod biomass, (**c**) small cladoceran biomass, (**d**) rotifers biomass, (**e**) *Daphnia* biomass, (**f**) chlorophyll a concentration. C: Control, F: Fish-added treatment, S: Shrimp-added treatment, FS: Fish–shrimp-added treatment. Values represent the mean ± SE (n = 6). Different letters above the bars indicate significant differences based on the results of one-way ANOVA.

**Figure 3 biology-12-00578-f003:**
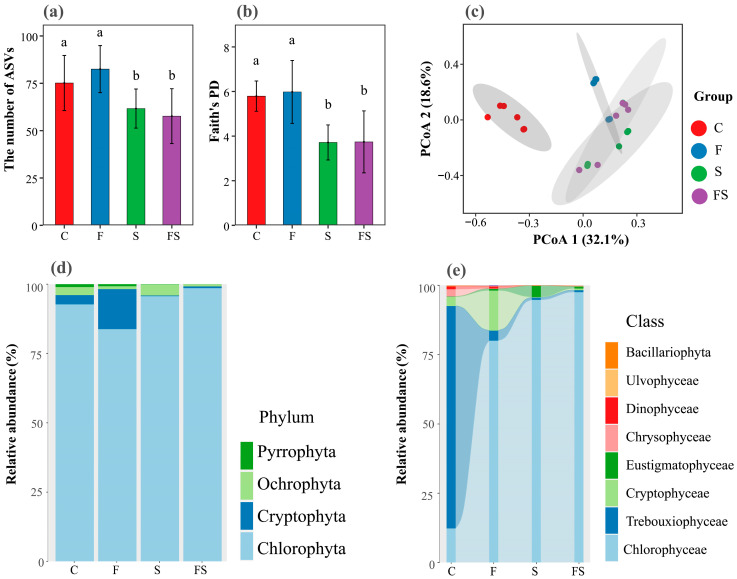
Phytoplankton alpha diversity and assemblage composition. (**a**) The number of phytoplankton amplicon sequence variants (ASVs); (**b**) Faith’s phylogenetic diversity (Faith’s PD); (**c**) Principal coordinate analysis (PCoA) plot of phytoplankton assemblage composition; Phytoplankton taxonomic distribution at the phylum (**d**) and class (**e**) levels. C: Control, F: Fish-added treatment, S: Shrimp-added treatment, FS: Fish–shrimp-added treatment. Different letters above the bars indicate significant differences according to one-way ANOVA analysis.

**Figure 4 biology-12-00578-f004:**
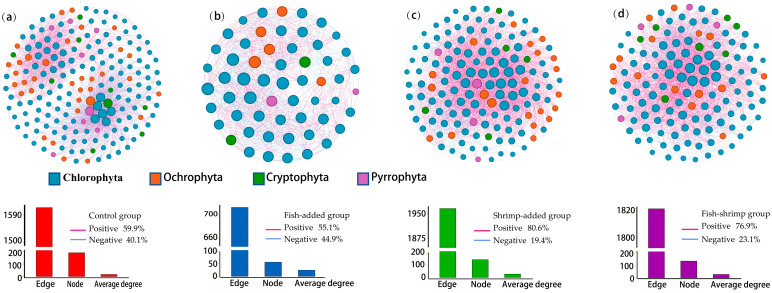
Difference in the network interactions of phytoplankton assemblages in the C (**a**), F (**b**), S (**c**), and FS (**d**) treatments. Nodes are colored according to phytoplankton phylum, while node size represents the degree of the node.

**Figure 5 biology-12-00578-f005:**
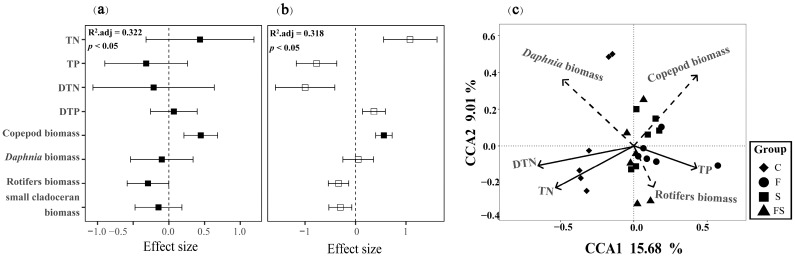
The outcome of a multiple linear regression in terms of the effects of predictor variables on the number of ASVs (**a**) and phylogenetic index diversity (**b**) (black solid lines indicate the mean ± SE, *p* < 0.05). Redundancy analysis of environmental indicators and biological factors in water bodies and changes in the composition of phytoplankton communities in the treatments (**c**) (solid black arrows indicate environmental indicators and dotted black arrows indicate biological factors. Shapes represent treatments).

**Figure 6 biology-12-00578-f006:**
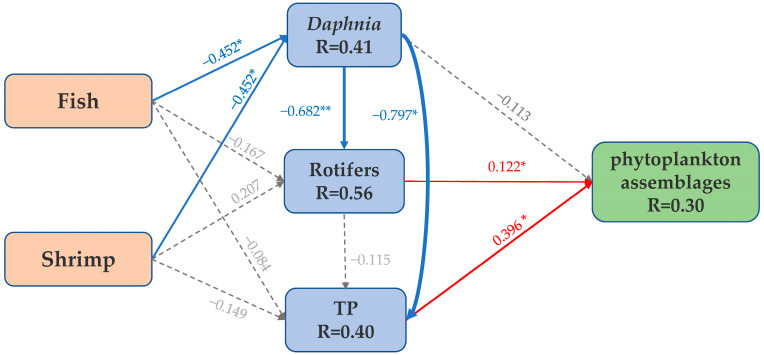
Structural Equation Model (SEM) of influencing factors in relation to changes in phytoplankton assemblages. Red solid lines represent positive correlations that passed the significance test, blue solid lines represent negative correlations that passed the significance test, gray dashed lines represent those that did not pass the significance test. ∗: *p* < 0.05; ∗∗: *p* < 0.01).

**Table 1 biology-12-00578-t001:** Generalized linear mixed model analysis of different treatments on the investigated factors.

Response Variable (Unit)	Variable	Estimates	SE	*z*-Value	*p*
TN (mg L^−1^)	I	2.56	0.15	16.6	**<0.001**
	F	−0.63	0.22	−2.91	**0.009**
	S	−0.46	0.22	−2.13	**0.046**
	F × S	0.69	0.31	2.24	**0.037**
TP (mg L^−1^)	I	0.04	0.01	4.89	**<0.001**
	F	0.04	0.01	3.43	**0.003**
	S	0.03	0.01	2.96	**0.008**
	F × S	−0.04	0.02	−2.73	**0.013**
Chl-a (μg L^−1^)	I	0.83	0.15	5.65	**<0.001**
	F	0.93	0.21	4.46	**<0.001**
	S	0.87	0.21	4.2	**<0.001**
	F × S	−1	0.29	−3.41	**0.003**
Zooplankton biomass (μg L^−1^)	I	2.81	0.08	35.85	**<0.001**
	S	−0.37	0.11	−3.3	**0.004**
	F × S	0.45	0.16	2.89	**0.009**
Copepod biomass (μg L^−1^)	I	1.69	0.24	7.05	**<0.001**
*Daphnia* biomass (μg L^−1^)	I	2.55	0.17	15.13	**<0.001**
	F	−2.55	0.24	−10.7	**<0.001**
	S	−2.55	0.24	−10.7	**<0.001**
	F × S	3.08	0.34	9.14	**<0.001**
Small cladoceran biomass (μg L^−1^)	I	2.26	0.13	18.02	**<0.001**
	S	−0.55	0.18	−3.12	**0.005**
	F × S	0.74	0.25	2.94	**0.008**

I, model intercept; F, fish effect; S, shrimp effect; F × S, interactive effect. Significant terms are in bold.

**Table 2 biology-12-00578-t002:** ADONIS and ANOSIM tests for phytoplankton assemblage composition in different treatments.

	ADONIS		ANOSIM	
	R	*p*	R	*p*
C vs. F	0.203	**0.004**	0.596	**0.001**
C vs. S	0.217	**0.007**	0.796	**0.004**
C vs. SF	0.216	**0.003**	0.743	**0.001**
F vs. S	0.175	**0.001**	0.496	**0.005**
F vs. SF	0.165	**0.003**	0.359	**0.004**
S vs. SF	0.093	0.397	0.002	0.417

*p*  < 0.05 terms in bold.

**Table 3 biology-12-00578-t003:** Topological parameters of the phytoplankton observation network and its associated stochastic network in different treatments.

Treatment	Observation Network							Random Network	
	Node	Edge	Positive Correlation	Modularity	Average Clustering Coefficient	Average Path Length	Network Diameter	Average Degree	Modularity (SD)
C	191	1612	59.9%	0.485	0.423	2.573	8	16.88	0.152 (0.008)
F	57	711	55.1%	0.18	0.227	1.926	3	24.95	0.111 (0.007)
S	143	1966	80.6%	0.182	0.214	1.95	4	27.5	0.110 (0.005)
SF	134	1820	76.9%	0.18	0.227	1.926	4	27.16	0.111 (0.007)

## Data Availability

The datasets generated during and/or analyzed during the present study are available from the corresponding author on reasonable request.

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
