# Peer review of "The Effects of Intraguild Predation on Phytoplankton Assemblage Composition and Diversity: A Mesocosm Experiment"

_biology, 2023, doi:10.3390/biology12040578_

Round 1
Reviewer 1 Report
Dear Authors,
The manuscript deals with an interesting topic of intraspecific food web chain in aquatic conditions. The authors did a great job according the implementation of the research. There were several typos and misspellings, please read the manuscript again and correct all of them.
Suggestions are as follows:
The authors may consider changing community to assemblages, due to the fact that in this case, the authors did not test interspecific correlations just intraspecific ones.
line47: Intraguild no need capital
line50: what does biological communities means? do we have non-biological communities too?
line 79-80: “previous studies” represent research, but at the end only one citation please correct it.
line86: why not provide a good description? many manuscripts studied phytoplankton assemblages used only microscopes and are still base and must-have-to-use publications… please be a little mild about these statements
introduction more information about biomanipulation and diversity co-occurrences, like those listed in: 10.3390/w11050929, 10.2478/s11756-018-0097-3, 10.3390/w14142149
line97: list author and date for Latin species names
line 114: use SI metric measures. ind. L-1
line121: at first mention, please use full names of species
lin128: same, use SI metrics.
line131: which part of the day was used for taking the samples?
fig1: use SI metrics. please include names for abbreviations
line120: duplicate word “treatment”
line246: significantly different from each other
figure 2c: C is ab significant based on this graph not bc
line297: PCoA not listed in the statistical analysis chapter in material and methods
line299: 32.1 and 18.6 % contribution seems too low
fig 4: too small, please make it bigger.
Otherwise, the manuscript is well-written and organized.
Reviewer 2 Report
Review for the paper "The effects of intraguild predation on phytoplankton community composition and diversity: A microcosm experiment" by Jun Da, Yilong Xi, Yunshan Cheng, Hu He, Yanru Liu, Huabing Li and Qinglong L. W submitted to "Biology".
General comment.
Autotriphic mucri-, nano- and picoplankton have been demonstrated to play several important roles in aquatic food webs, e.g. as primary producers, thus controlling the efficiency of carbon cycling in any ecosystem. The role of zooplankton as phytoplankton consumers has been extensively studied over the previous two decades, across a variety of aquatic habitats (freshwater to oceanic environments) and geographic regions (polar to tropical waters), thereby showing the general importance of zooplankton grazing in the freshwaters. Available data regarding the impact of zooplankton grazing on phytoplankton diversity remain limited, particularly in temperate and subtropical regions, in which the levels of phytoplankton biomass and nutrient concentrations are expected to vary considerably in time and space. The authors investigated how intraguild predation could affect phytoplankton assemblages by applying a model study on experimental conditions. They used environmental DNA high-throughput sequencing and revealed several effects related with phytoplankton structure in responses to different treatments (addition of two predator species). Network indicated that the two predator taxa had contrasting impacts on the total phytoplankton diversity. The study expands current knowledge regarding the influence of grazing on the structure and functioning of aquatic ecosystems and may be interesting for an international audience. Standard methods to collect samples and to treat the data were used in the study. The main results are illustrated with relevant Figures and Tables. The Discussion is comprehensive and focused on the main findings. Statistical methods are adequate and correctly used. I have some minor suggestions to improve the article.
Specific remarks.
L3. Consider replacing "A microcosm experiment" with "A mesocosm experiment".
L58-59. Consider replacing " the feeding range of small zooplankton is smaller than that of large zooplankton [8]," with "small zooplankton had higher prey selectivity than large zooplankton [8],".
L114. Consider replacing "daphnia" with "Daphnia" (Daphnia in Italic).
L118-120. Incubation conditions for each meosocosm experiment must be clearly described (temperature, light regime, nutrients etc). Also, experimental conditions must be carefully presented here or in another part of the Material and methods.
L121. Indicate the full Latin names for P. fulvidraco and E. modestus and provide a short description regarding the trophic position and role of these two predators in Lake Taihu.
L146. The authors must note that they used dry zooplankton biomass.
L189-192. Please, provide a detailed description of GLMM (assumptions, response and predictor variables).
Table 1. The title of the column 'coefficient' appears to be not appropriate. May be better using 'Variable'?
L245. The authors noticed that the addition of P. fulvidraco and E. modestus influenced the total biomass of phytoplankton and zooplankton. However, they did not determine phytoplankton biomass directly but used chlorophyll a concentration as a measure of phytoplankton biomass. Therefore, I recommend replacing "phytoplankton" with "Chl-a" or the authors should clearly indicate in the Material and methods that Chl-a was the equivalent of phytoplankton biomass.
L279-290 and Fig. 3, 4, L477-478. The Latin names of the phyla and families must be in ordinary font according to the International nomenclature.
L342. Consider replacing "rotifer" with "rotifers".
Fig. 5. "Daphnia" must be in italics.
Fig. 6. Consider replacing "Rotifer" with "Rotifers" or "Rotifera".
L411. Consider replacing "phytoplankton growth" with "phytoplankton diversity and composition" or "Chl-a biomass".
L447. Consider replacing "use" with "using".
L453. Consider replacing "nutrient salinity" with "nutrient concentrations".
L464-466. The authors have found an interesting result regarding the positive relationships between phytoplankton diversity and copepod biomass (L344-344). However, they did not interpret this relationship. Provide the possible biological mechanisms explaining an increase in phytoplankton diversity in response to the increasing copepod biomass.
Literature cited. In some places the Latin names of the taxa are in ordinary font while these must be in italics (e.g. L549, 555 and below). I suggest carefully check the References and correct the Latin names.
